# The Evolution of Artificial Intelligence in Medical Imaging: From Computer Science to Machine and Deep Learning

**DOI:** 10.3390/cancers16213702

**Published:** 2024-11-01

**Authors:** Michele Avanzo, Joseph Stancanello, Giovanni Pirrone, Annalisa Drigo, Alessandra Retico

**Affiliations:** 1Medical Physics Department, Centro di Riferimento Oncologico di Aviano (CRO) IRCCS, 33081 Aviano, Italy; giovanni.pirrone@cro.it (G.P.); adrigo@cro.it (A.D.); 2Elekta SA, 92100 Boulogne-Billancourt, France; joseph.stancanello@elekta.com; 3National Institute for Nuclear Physics (INFN), Pisa Division, 56127 Pisa, Italy; alessandra.retico@pi.infn.it

**Keywords:** artificial intelligence, medical imaging, neural networks, machine learning, deep learning

## Abstract

Artificial intelligence, now one of the most promising frontiers of medicine, has a long and tumultuous history punctuated by successes and failures. One of its successes was its application to medical images. We reconstruct the timeline of the advancements in this field, from its origins in the 1940s before crossing medical images to early applications of machine learning to radiology, to the present era where artificial intelligence is revolutionizing radiology.

## 1. Introduction

Artificial intelligence (AI) permeated medicine slowly but steadily, at first through seminal works and then with the first commercial systems, until the present day, when AI now represents one of the most promising frontiers of medicine. Researchers have shown that AI can perform a wide range of tasks in medical imaging [1], with recent prospective clinical trials indicating that AI achieves performance levels comparable to humans in diagnostic tasks [2]. However, the widespread adoption of AI in medicine remains challenging, as it requires ensuring its safe and ethical use [3,4].

In this narrative review, we will describe the history of the development of AI, from the first conceptualizations of learning machines in the 1940s to the modern refinements regarding neural networks, which allow for the successful usage of AI in most, if not all, human disciplines. We will also describe the advancements in the use of AI in medicine, from the first expert systems to modern applications of neural networks in imaging, in particular for oncological applications. The review consists of three main sections: in the first, we describe the works of the pioneers of AI in the 20th century, a time when AI was not interested in nor capable of analyzing medical images. The second section describes an era when the first AI-based classifications of imaging findings were attempted, with the first widely used machine learning (ML) algorithms. In the present era, medical imaging is being transformed by the endless possibilities offered by a large spectrum of neural network architectures.

## 2. AI Before Meeting Medical Imaging: From the Origins to Expert Systems

AI is an umbrella term covering a wide spectrum of technologies aiming to give machines or computers the ability to perform human-like cognitive functions such as learning, problem-solving, and decision-making. At its beginning, the goal of AI was to imitate the human mind or, in the words of Frank Rosemblatt, to make a computer “able to walk, talk, see, write, reproduce itself and be conscious of its existence” [5]. However, it was soon understood that AI could achieve better results at well-defined, specific tasks such as playing checkers [6], to the level of surpassing the performance of humans, e.g., the computer Deep Blue defeating former world chess champion Garry Kasparov in 1997 [7]. In this section, we describe this early phase.

### 2.1. Prehistory of AI

The idea of inanimate objects being able to complete tasks that are usually performed by humans and require “intelligence” dates back to ancient times [8]. The history of AI started with a group of great visionaries and scientists in the 1900s, including Alan Turing (London, 1912–Manchester, 1954, Figure 1a), one of the fathers of modern computers. He devised an abstract computer called the Turing machine, a concept of paramount importance in modern informatics, as any modern computer device is thought to be a special case of a Turing machine [9]. He also worked on a device, the Bombe, at Bletchley Park (75 km northwest of London), which involved iteratively reducing the space of solutions from a set of message transcriptions to discover the decryption key of enemy messages during World War II [10]. This process had some resemblance to ML, which hypotheses a model from a set of observations [11]. A timeline of the origin and development of AI, starting from the Turing machine to the triumph of artificial neural networks (ANNs), is provided in Figure 2.

In a public lecture held in 1947, Turing first mentioned the concept of a “machine that can learn from experience” [12] and posed the question “can machine think?” in his seminal paper entitled “Computing machinery and intelligence” [13]. In this paper, the “imitation game”, also referred to as the Turing test, was proposed to define if a machine can think. In this test, a human interrogates another human and the machine alternatively. If it is not possible to distinguish the machine from the human based on the answers then the machine passes the test and is considered able to think.

Turing discussed strategies for achieving a thinking machine by programming and learning. He likened the learning process to that of a child being educated by an adult who provides positive and negative examples [11,14]. From its very beginning, different branches of AI emerged. Symbolic AI searches for the proper rule (e.g., IF-THEN) to apply to the problem at hand by testing/simulating all the possible rules, like in a chess game, without training [15,16]. On the other hand, ML is characterized by a training phase, where the machine analyses a set of data, builds a model, and measures model performance through a function called goal or cost function [17]. The term ML was introduced by Arthur L. Samuel (Emporia, USA, 1901–Stanford, USA, 1990), who developed the first machine able to learn to play checkers [6]. The dawn of AI is considered the summer conference at Dartmouth College (Hanover, NH, USA) in 1956 [18]. At the meeting, “artificial intelligence” was defined by John McCarthy (Boston, USA, 1927–Stanford, USA, 2011) as “the science and engineering of making intelligent machines”. This definition, as well as the implicit definition of AI in the imitation game, escapes the cumbersome issue of defining what intelligence is [19], making the goals and boundaries of the science of AI blurry. For instance, in the early years of AI, research clearly targeted computers that could have performance comparable with those of the human mind (”strong AI”). In later years, the AI community shifted its aim to more limited realistic tasks, like solving practical problems and carrying out individual cognitive functions (“weak AI”) [19].

### 2.2. Neural Networks

Neural networks were first conceived in 1943 by Warren S. McCullough (Orange, NJ, USA, 1898–Cambridge, MA USA, 1969), a neuroscientist, and Walter Pitts (Detroit, USA, 1923–USA, 1969), a logician, as an abstract model to describe the functioning of the brain [20]. It consisted of a network of units (nodes) that simulate the brain cells, the neurons, which could receive a limited number of binary inputs and send a binary output to the environment or other neurons [21,22]. This early model was not designed to be able to learn [23], as the weights of its units and signals were fixed, in contrast to modern ML networks, which have learnable weights [24]. In 1949, the psychologist Donald O. Hebb (Chester, Canada 1904–Chester, Canada, 1985) introduced the first rule for self-organized learning: “any two cells or systems of cells that are repeatedly active at the same time will tend to become associated: so that activity in one facilitates activity in the other” [25], meaning that the weights of connections are increased when frequently used simultaneously [26]. Inspired by these works, Marvin L. Minsky and Dean Edmonds built an analog neural network machine called “stochastic neural-analog reinforcement calculator” (SNARC), which could determine a way out of a maze [27].

Shortly afterward, a learning neural network machine, the perceptron (Figure 3), was developed by the psychologist Frank Rosenblatt (New Rochelle, NY, USA, 1928–Chesapeake Bay, USA, 1971) [28]. Since it was built for the classification of a binary image generated using a camera, it can be considered the first application of AI to images [29]. It used a Heaviside step as an activation function, which converts an analog signal into a digital output [30]. The learning was accomplished by a delta rule, where delta is simply the difference in the network output and the true value, and an incorrect response is used to modify the weights of the connections towards the correct pattern of prediction [30]. If multiple units are organized in a layer to be able to produce multiple outputs from the same input, we obtain a structure that is today called single-layer artificial neural networks (ANNs) [5]. In 1960, the adaptive linear neuron (ADALINE) used the weighted sum of the inputs to adjust weights [31] so that it could also estimate how much the answer was correct or incorrect in a classification problem [32].

### 2.3. Supervised and Unsupervised ML

ML is used to explore data (‘data mining’) to identify variables of interest and uncover useful correlations and patterns without any predefined hypothesis to test. In this sense, ML operates inversely to traditional statistical approaches, which begin with a hypothesis [33]. The most common approach is supervised learning, where the system uses training data with corresponding ground truth labels to learn how to predict these labels [34]. In unsupervised ML, the training data have no ground truth labels, and the ML learns patterns or relationships in the data, resulting in data-driven solutions for dimensionality reduction, data partitioning, and the detection of outliers. To the first category belongs the principal component analysis, PCA [35], which uses an orthogonal linear transformation to convert the data into a new coordinate system to perform data dimension reduction [36]. PCA is useful when a high number of variables may cause ML models to overfit. Overfitting occurs when a model memorizes the training examples but performs poorly on independent test sets due to a lack of generalization capability [34].

### 2.4. First Applications of AI to Medicine: Expert Systems

In 1969, Minsky and Papert proved [37] that a single-layer ANN was not able to solve classification problems where the separation function is nonlinear [38]. Given their undisputed authority in the field, the interest and funding in neural networks decreased until the early 1980s, leading to the “first AI winter”. During this era, researchers tried to develop systems that could operate in narrower areas. The idea initially came to Edward A. Feigenbaum (Weehawken, NJ, USA, 1936–), who became interested in creating models of the thinking of scientists, especially the processes of empirical induction by which hypotheses and theories were inferred from knowledge in a specific field [39]. As a result, he developed expert systems, computer programs that make a decision such as a medical diagnosis using a knowledge database, and a set of IF-THEN rules [21]. The first was the DENDRAL [40], which could derive molecular structure from mass spectrometry data by using an extensive set of rules [27]. One of the first prototypes to demonstrate the feasibility of applying AI to medicine was CASNET, a software to provide support on diagnosis and treatment recommendations for glaucoma [41]. MYCIN was designed to provide disease identification and antibiotic treatment based on an extensive set of rules and patient data. It was superseded by EMYCIN and the more general purposing INTERNIST-1 [42]. These systems aiming at supporting the clinician’s decision are called computer decision support systems (CDSSs).

Expert systems were limited by poor performance in areas that cannot be easily represented by logic rules, such as detecting objects with significant variability in images. In addition, they cannot learn from new data and update their rules accordingly, resulting in a lack of adaptability [10]. For these reasons, in the 1990s, the interest shifted to ML, as the larger availability of microcomputers coincided with the development of new popular ML algorithms such as SVMs and ensemble decision trees [43].

## 3. Early Applications of AI to Imaging: Classical ML and ANNs

Gwilym S. Lodwick, in 1963, calculated the probability of bone tumor diagnosis with good accuracy based on observations such as the lesion’s location relative to the physis and whether it was in a long or flat bone [44] using an ML algorithm, specifically the Bayes rule [45]. This early attempt can be considered the first application of ML to medical images. Due to its low computational cost, this approach—by extracting descriptive features from images and then analyzing them with ML models—dominated the AI field for many years until the advent of deep learning.

### 3.1. Decision Tree Learning

A decision tree is a set of rules for partitioning data according to their attributes or features (Figure 4a). This is a very intuitive process. In fact, the first classification tree is the Porphyrian tree, a device by the 3rd-century Greek philosopher Porphyry (Tyre, Roman Empire, present-day Lebanon, 234–Rome, 305) to classify living beings. Decision trees can be combined with ML in decision tree learning, where one or more decision trees are grown to create partitions of data according to rules based on the data features for classification or regression of data.

The first attempt at decision tree methodology can likely be traced to the mid-1950s with the work of the statistician William A. Belson. He aimed to predict the degree of knowledge viewers had about the “Bon Voyage” television broadcast by using demographic variables such as occupational and educational levels [46], overcoming the limitations of linear regression [47,48,49]. Later, J.N. Morgan and J.A. Sonquist [50] proposed what is now considered the first decision tree method, popularized thanks to the AID computer program [49]. Impurity is a measure of the class mix of a subset, and splits are chosen so that the decrease in impurity is maximized. Currently, the Gini index [51] is the preferred method for measuring impurity; it represents the probability of a randomly chosen element being incorrectly classified so that a value of zero means a completely pure partition. The Classification And Regression Trees−CART by Leo Breiman also used pruning, a process that reduces the tree size to avoid overfitting [52,53]. It is still largely used in imaging analysis due to its intuitiveness and ease of use, e.g., it could classify tumor histology from image descriptions in MRI [54].

### 3.2. Support Vector Machines and Other Traditional ML Approaches

The aim of Support Vector Machines (SVMs) is to determine a hyperplane in the *n*-dimensional space of attributes that separates data into two or more classes for the purpose of classification. The searched hyperplane is such that the minimum distance from it to the convex hull (i.e., the minimum enclosing a set of points [55]) of classes is maximal. This idea was first proposed by Vladimir Vapnik and Alexey Chervonenkis in 1964 [56]. A few years later, Corinna Cortes and Vladimir Vapnik [57] proposed the first soft-margin SVM. The latter allows the inclusion of a certain number of misclassified data while keeping the margin as wide as possible so that other points can still be classified correctly. SVMs can also perform nonlinear classification using the “kernel trick”, a mapping to higher dimensional feature space using proper transformations (e.g., polynomial functions, radial basis functions). SVMs are one of the most frequently used ML approaches in medical data analysis [58], and they have been found, for instance, to provide accurate results for the classification of prostate cancer from multiparametric MRI [59]. SVM classifiers were also widely used in the analysis of neuroimaging data, e.g., in the study of neurodevelopment disorders [60] and of neurodegeneration [61]. An example of SVM classification is provided in Figure 4b.

Naïve Bayes learning, another popular ML approach, involves constructing the probability of assigning a class to a vector of features based on Bayes’ theorem and then assigning the class with the maximum probability [62,63]. The K-nearest neighbors (KNNs) algorithm was introduced by T. Cover and P. Hart in 1967 [64]. Its formulation appears to have been made by E. Fix and J.L. Hodges in a research project carried out for the United States armed forces, which introduced discriminant analysis, a non-parametric classification method [65]. They investigated a rule that might be called the KNN rule, which assigns to an unclassified sample point the classification of the nearest of a set of previously classified points.

Traditional machine learning (ML) models analyze input data structured as vectors of attributes, also known as variables, descriptors, or features. These features can be either semantic (e.g., “spiculated lesion”) or agnostic (quantitative) [66]. Coding a problem in terms of a feature vector can lead to an extremely large number of features depending on the complexity of the problem to address, increasing the risk of overfitting. Feature selection is a process to determine a subset of features such as all the features in the subset are relevant to the target concept, and no feature is redundant [67,68]. A feature is considered redundant when adding it on top of the others will not provide additional information; for instance, if two features are correlated, these are redundant to each other [69]. Feature selection may be considered an application of Ockham’s razor to ML. According to Ockham’s razor principle, attributed to the 14th-century English logician William of Ockham (Ockham, England, 1285–Munich, Bavaria, 1347), given two hypotheses consistent with the observed data, the simpler one (i.e., the ML model using the lower number of features), should be preferred [70]. Depending on the type of data, feature selection can be classified as supervised, semi-supervised, or unsupervised [69]. There are three main classes of feature selection methods: (i) embedded feature selection, where ML includes the choice of the optimal subset; (ii) filtering, where features are discarded or passed to the learning phase according to their relevance; and (iii) wrapping, which requires evaluating the accuracy of a specific ML model on different feature subsets for choice of the optimal one [67]. “Tuning” is the task of finding optimal hyperparameters for a learning algorithm for a considered dataset. For instance, decision trees have several hyperparameters that may influence their performance, such as the maximum depth of the tree and the minimum number of samples at a leaf node [71]. Early attempts for parameter optimization include the introduction of the Akaike information criteria for model selection [72]. More recent strategies include grid search, in which all parameter space is discretized and searched, and random search [73], in which values are drawn randomly from a specified hyperparameter space, which is more efficient, especially for ANNs [71].

### 3.3. First Uses of Neural Networks for Image Recognition

To address the criticism of M. Minsky and S. Papert [37] and enable neural networks to solve nonlinearly separable problems, many additional layers of neuron-like units must be placed between input and output layers, leading to multilayer ANNs. The first work proposing multilayer perceptrons was published in 1965 by Ivakhnenko and Lapa [71]. A multilayered neural network was proposed in 1980 by Fukushima called “Neocognitron” [74], which was used for image recognition [75], and included multiple convolutional layers to extract image features of increasing complexity. These intermediate layers are called *hidden layers* [21] and multilayer architectures of neural networks are called “deep”. Hence, the term “*deep learning*” (*DL*) was coined by R. Dechter [76]. The difference between single-layer and multilayer ANNs is shown in Figure 5.

In a DL neural network, training is performed by updating all the weights simultaneously in the opposite direction to a vector that indicates by the change in the error if weights are changed by a small amount to search a minimum in the error in response. This method is called “standard gradient descent” [77], and one of its limitations was that it made tasks such as image recognition too computationally expensive.

The invention of the backpropagation learning algorithm in the mid-1980s [78] improved significantly the efficiency of training of neural networks. The back-propagation equations provide us with a way of computing the gradient of the cost function starting from the final layer [79]. Then, the backpropagation equation can be applied repeatedly to propagate gradients through all modules all the way to the input layer by using the chain rule of derivatives to estimate how the cost varies with earlier weights [80]. This learning rule and its variants enabled the use of neural networks in many hard medical diagnostic tasks [81].

The introduction of the rectifier function or ReLu (rectified linear unit), an activation function, which is zero if the input is lower than the threshold and is equal to the input otherwise, helped reduce the risk of vanishing/exploding gradients [82] and is the most used activation function as of today.

Despite this progress, AI entered its second winter at the beginning of the 1990s,, which involved the use of neural networks, to the point that, at a prominent conference, it was noted that the term “neural networks” in a manuscript title was negatively correlated with acceptance [83]. This new AI winter was partly due to the vanishing/exploding gradient problem of DL, which is the exponential increase or decrease in the backpropagated gradient of the weights in a deep neural network.

### 3.4. Ensemble Machine Learning

During the second winter of AI in the 1990s, while neural networks saw a decrease in interest, AI research was focused on other ML techniques, such as SVM and decision trees, and on ways to improve their accuracy. Significant improvements in the accuracy of decision trees arise from growing an ensemble of trees and subsequently aggregating their predictions [84]. In bagging aggregation, to grow each tree, a random selection is made from the examples in the training set [85], using a resampling technique approach called “bootstrap” [86]. The bagging aggregation averages over the versions when predicting a numerical outcome and does a plurality vote when predicting a class [87]. Random forest, proposed in 1995 by Tin K. Ho [88], is an example of this approach. In boosting aggregation [89], the distribution of examples is filtered in such a way as to force the weak learning algorithm to focus on the harder-to-learn parts of the distribution. The popular Adaboost (from ‘adaptive boosting’) reweights individual observations in subsequent samples and is well suited for imbalanced datasets [90].

### 3.5. ML Applications to Medical Imaging: CAD and Radiomics

Computer-aided detection or diagnosis (CAD) systems assist clinicians by analyzing medical images and highlighting potential lesions or suggesting diagnoses [91,92]. Early CAD systems used hand-crafted image features, which were introduced into rule-based algorithms to produce an index (e.g., a probability of malignancy) to be used for diagnosis [93]. Features could include spiculations, roughness of margins, and perimeter-to-area ratio for distinguishing malignant breast lesions [94] or lung disease in radiography [95].

The first commercial CAD system was the ImageChecker M1000 (R2 Technology, Los Altos, CA, USA), which received US Food and Drug Administration−FDA approval in 1998 [96] and provided the likelihood of malignant lesions according to the presence of clusters of bright spots and spiculated masses, also highlighting regions at risk of malignancy [93]. Other breast CAD systems were proposed for breast magnetic resonance imaging: CADstream (Merge Healthcare Inc., Chicago, IL, USA) and DynaCAD for breast (Invivo, Gainesville, FL, USA) [93]. Early CAD systems exhibited lower specificity and positive predictive value compared to double reading by radiologists, rendering the sole use of CAD not advisable [97].

Textural features, first introduced by Robert.M. Haralick and coworkers in 1973 [98], began to be used for the quantitative analysis of texture with ML for pattern recognition on computed tomography [99]. The term “radiomics” first appeared in 2010 [100,101], combining the terms “radio”, referring to radiological sciences, and the suffix “omics”, often used in biology (e.g., genomics, transcriptomics, and proteomics), to emphasize a research field encompassing the entire view of a system by mining a large amount of data [102]. Radiomics focused on investigating the tumor phenotype in imaging for building prognostic and predictive models [103], in particular for oncological applications [14]. Thus, it has largely contributed to the idea that ML can be applied to quantitatively analyze images [104]. An array of ML techniques is currently used for radiomics, including SVM and ensemble decision trees [1]. The radiomic approach, complemented by ML, has been largely implemented in a large variety of studies devoted to the identification of imaging-based biomarkers of disease severity assessment or staging and patient’s outcome or risk for side effects [105,106,107,108,109]. The scientific community is still investigating the robustness and reproducibility of radiomics features and their dependence on image acquisition systems and parameters across different modalities [110,111].

## 4. The Era of Deep Learning in Medical Imaging

In 1989, Yann LeCun introduced the concept of a convolutional neural network (CNN) to recognize handwritten digits, paving the way for the use of deep neural networks in imaging [112]. CNNs have layers that perform a convolution operation with a kernel that acts as a filter, e.g., a Sobel filter, whose effect is shown in Figure 1b,c. The numerical values of the kernels that operate image filtering are not fixed a-priory; they are learned from data and set during the training phase [93]. Unlike the traditional ML approach, the DL does not require the extraction of meaningful features to describe the image. The performance of CNNs can be boosted by artificially augmenting the dataset by affine transformations of the images, like translation, scaling, squeezing, and shearing, a strategy that reduces overfitting [83]. Further improvements were achieved with the introduction of specialized layers, i.e., stacks of neural units that perform a particular task within the DL architecture like dropout [113], pooling, and fully connected layers [93], allowing an endless spectrum of configurations for the most diverse tasks. In 2012, a deep CNN developed at the University of Toronto demonstrated excellent performance at the ImageNet Large Scale Visual Recognition Challenge in 1.2 million high-resolution images of 1000 classes [114]. Competitions or challenges have become a major driver of progress in AI [4,115]. Autoencoders and Convolutional Autoencoders [116] are used for dimensionality reduction, data and image denoising, and uncovering hidden patterns in unlabeled data. Additionally, sparse autoencoders can generate extra useful features [117].

Consequently, it is now clear that DL can be designed for almost any many domains of science, business, and government to perform as diverse as image, signal, and sound recognition, transformation, or production.

### 4.1. Medical Images Classification with Deep Learning Models

Neural networks began to be used in CAD by M.L. Giger and coworkers in radiographic images [91,118,119], e.g., in lung [120] and breast [92] investigations. In the 1990s [118] researchers started to use CNNs to identify lung nodules [121,122], and detect micro-calcifications in mammography [123]. Since the first attempts, CNNs have demonstrated a great potential to solve a large variety of classification tasks; thus, they have been implemented across many pathologies and imaging modalities [124]. The CardioAI CAD system was one of the first neural network-based commercial systems for analyzing cardiac magnetic resonance images [125]. Moreover, AI can integrate data from different modalities, an approach termed multimodal AI or multimodal data fusion, which can, for instance, be used to diagnose cancer using both imaging and patient EHR data. This task can be accomplished by designing neural networks that accept multiple data, resulting in multidimensional analysis [126]. Recently, the You Only Look Once (YOLO) neural network architecture, initially designed for real-time object detection, has been under investigation for polyp detection in colonoscopy [127] and identifying dermoscopic and cardiovascular anomalies [128].

Among the CNN-based systems approved for clinical use, ENDOANGEL (Wuhan EndoAngel Medical Technology Company, Wuhan, China) can provide an objective assessment of bowel preparation every 30 s during the withdrawal phase of a colonoscopy [129]. The CNN-based system can also analyze images to predict overall survival and occurrence of distant metastases [130].

These DL models are characterized by a very large number of free parameters that must be set during the training phase, making network training from scratch computationally intensive. In transfer learning [131,132], the knowledge acquired in one domain is transferred to a different one, much like a person using their guitar-playing skills to learn the piano [133]. This allows a learner in one domain (e.g., radiographs) to leverage information previously acquired by models such as the Visual Geometry Group (VGG) and Residual Network (ResNet), which were trained on a related domain (e.g., images of common objects).

### 4.2. Segmentation with Deep Learning Models

AI can be used in medical image analysis to automatically subdivide an image into several regions based on the similarity or difference between regions, thus performing segmentation of soft tissues and lesions, a tedious task if performed manually. Initially, segmentation was pursued using semiautomated approaches, such as edge-, region-, or threshold-based segmentation. For instance, Sobel filters were applied to enhance and detect lesion borders. However, these methods are highly sensitive to noise and image contrast, limiting their effectiveness [134]. Then, automated segmentation was attempted by using unsupervised [135] or, supervised ML [136]. Another approach consists of calculating radiomic features in the neighborhood of pixels and then classifying them using ML [137]. Image segmentation was made more efficient by the U-Net CNN [138] currently used for a large variety of segmentation tasks across different image modalities [139,140,141,142,143,144]. It consists of an encoder branch where the input layer is followed by several convolutional and pooling layers, as in a CNN architecture for image classification; then, a symmetric decoder branch allows obtaining a segmentation mask with the same dimension of the input image. The U-Net’s peculiar skip connections bridge the encoder and decoder, directly transferring detailed spatial information to the upsampling path enabling precise object locations in the final segmentation masks [145]. In 2016, Ö. Çiçek and coworkers [146] presented a modified three-dimensional version of the original U-Net (3D U-Net) for volumetric segmentation.

### 4.3. Medical Image Synthesis: Generative Models

Generative Adversarial Networks (GANs) represent a DL architecture capable of generating new and realistic images by training from a dataset of images, video, or other types of data [147]. GANs include two DL networks, a generator, and a discriminator that are trained in an adversarial way, the target goal being to train the generator to produce an image realistic enough to induce the discriminator into classifying it as real. By using GANs it is possible to generate images from other images or from a text string, for instance. One of the most recent architectures for image generation is stable diffusion, which can produce high-quality images from text or text-conditional images, e.g., “basal cell carcinoma” in a dermoscopic image [148].

The use of generative models has proven to be valuable in medical imaging applications, including data synthesis or augmentation [149,150]. An example could be the generation of pseudo-healthy images, images that visualize a negative image of a patient that is being examined in order to facilitate lesion detection [151]. Virtual patient cohorts can be synthesized to perform virtual clinical trials for testing test new drugs, therapies, or diagnostic interventions, thus reducing the cost of clinical trials on humans [152].

Other applications include image denoising and artifact removal [153], image translation between different modalities [154], and multi-site data harmonization [155]. Fast AI-based image reconstruction also emerged to allow real-time MRI imaging [156]. GAN-based MRI image reconstruction particularly excels in capturing fine textures [157]. CNNs have also been applied to rigid [158] and deformable image registration [159], which is necessary to precisely track the absorbed dose in radiotherapy treatments at the voxel level [160]. A new promising neural architecture, the neural fields, can perform efficiently any of the above tasks by parameterizing the physical properties of images [161].

### 4.4. From Natural Language Processing to Large Language Models

Recurrent neural networks (RNNs) process an input sequence one element at a time, maintaining in their hidden units a ‘state vector’ that implicitly contains information about the history of all the past elements of the sequence. RNNs can predict the next character in a text or the next word in a sequence [162], making them useful for speech and language tasks. However, their training is challenging because the backpropagated gradients can either grow or shrink at each time step. Over many time steps, this can lead to gradients that either explode or vanish [163]. In 1997, long short-term memory (LSTM) RNNs were invented [164], which solve the problem of vanishing gradients for sequences of symbols by including a forget gate, which allows the LSTM to reset its state [165,166]. This subfield of AI aiming at developing the computer’s abilities to understand or generate human language is called natural language processing (NLP) [167]. There has been a surge of research in NLP diagnostic models from structured or unstructured electronic health records (EHR) [168,169,170]. In 2007, IBM introduced Watson, a powerful NLP software which, in 2017, was instrumental in identifying new RNA-binding proteins linked to amyotrophic lateral sclerosis [125].

A breakthrough in this field was the Transformer DL architecture, which, by employing the self-attention mechanism [171] showed excellent capabilities in managing dependencies between distant elements in an input sequence and in exploiting parallel processing to reduce execution times. Transformers are the basic components of Large Language Models (LLM), such as the Generative Pretrained Transformers (GPT) by OpenAI or the Bidirectional Encoder Representations from Transformers (BERT) by Google. These are trained on a large amount of data from the web and are able to generate text to make translation, summarization, and complete sentences, and also the production of creative content in domains specified by the users. LLM can be used as a decision support system that recommends appropriate imaging from a patient’s symptoms and history [172]. Recently, GPT4, a new version of the ChatGPT by OpenAI, a generative LLM that can generate human-like answers, was released. GPT4 can analyze images, implying that, if successfully applied to radiology, it could writes diagnoses from images [172] and can act as a virtual assistant to the radiologist [173].

A transformer-based encoder-decoder model analyzing chest radiograph images to produce radiology report text was assessed by comparing its generated reports with those generated by radiologists [174].

Beyond the automated image interpretation tasks, since the public availability of the ChatGPT chatbot at the end of 2022, its potential use, for example, in assisting clinicians in the generation of context-aware descriptions for reporting tasks, became apparent [172]. Similar uses entail a series of implications that are much debated in the community [175]. Moreover, LLMs can aid in patients comprehending their reports by summarizing information at any reading level and in the patient’s preferred language [173]. The architectures initially developed to understand and generate text have soon been adapted for other tasks in several domains, including computer vision. Vision Transformers (ViTs) [176] are a variant of transformers specifically designed for computer vision tasks, such as image classification, object detection, and image generation. Instead of processing sequences of word tokens (i.e., elements of textual data such as words and punctuation marks) as they do in NLP tasks, ViTs process image patches to accomplish relevant tasks in medical image analysis, such as lesion detection, image segmentation, registration, and classification [171]. The image generation processes operated by GANs could be further enhanced by implementing the attention mechanisms [167]. Combining ViTs for analyzing diagnostic images with LLMs for clinical report interpretation could result in a comprehensive image-based decision support tool. An extremely appealing use of Transformers is their potential to handle multimodal input data. This capability was demonstrated in the work by Akbari et al. [177], where a transformer-based architecture, the Video-Audio-Text Transformer, was developed to integrate images, audio, and text.

The possibility of implementing modality-specific embedding to convert the entries of each modality to processable information for a transformer-like architecture opens the possibility of analyzing in a single framework heterogeneous data types, which would be extremely relevant for medical applications where complementary information is encoded in textual clinical reports and tests, medical imaging, genetic and phenotypic information [152].

### 4.5. Foundational Models

A limitation of the AI-based tools discussed so far is their ‘narrow scope’, as they are typically designed to detect specific image abnormalities [178]. In contrast, models like GPT-4 are pre-trained on vast and diverse datasets that encompass text, audio, and images, allowing for broader applications. These are referred to as ‘foundation models’ because they can be fine-tuned for specific tasks using transfer learning, serving as the foundation for models capable of addressing specialized tasks [179]. This is a radical shift from previous artificial intelligence tools that were designed to solve specific tasks [180]. Recently, a foundation model trained on ImageNet, a large database of natural images (https://www.image-net.org/), was fine-tuned to generate realistic chest x-ray images based on prompts from the user [181]. A foundation model, after fine-tuning for pathology, was capable of nuclear segmentation, primary and metastatic cancer detection, cancer grading, and sub-typing, outperforming previous state-of-the-art models [182].

Since foundation models can perform various tasks across diverse domains, they can be adapted through in-context learning—introduced in 2020 with the GPT-3 language model—where the model learns from user-provided text explanations (or ‘prompts’) with a few examples [180]. In this way, models can adapt to new distributions of data on the fly using limited data, whereas traditional AI models need extensive retraining on a new dataset. A hospital, for instance, can teach a model to interpret X-rays from a brand-new scanner simply by providing prompts that show a small set of examples [180].

GPT-4 was instructed by users who constructed textual prompts from 25 CT radiology reports. The GPT-4 was then able to perform various tasks on unseen reports, such as extracting lesion parameters and identifying metastatic disease with high accuracy [183]. In this way, foundation models can circumvent the problem of data scarcity in medical imaging [184]. Another mechanism for this purpose is self-supervised learning, where the models build data representations by solving pretext tasks. Pretext tasks are tasks whose outcome is not of interest, such as image colorization, but result in the model learning representations of input images, improving its generalizability [185].

## 5. Open Challenges and Pathways for AI in Medical Imaging

In 2017, for the first time in history, DeepMind’s AlphaGo, a self-trained system based on a deep neural network, beat the world champion in arguably the most complex board game (called “Go”), thus achieving superhuman performance [186]. It is no wonder, then, that AI has achieved physician-level accuracy in a broad variety of diagnostic tasks, including image recognition [2], segmentation, and generation [187,188]. The development of graphics processing units (GPUs) and cloud computing has provided the significant computational power required to train DL models on large datasets. Additionally, they have made it possible to perform AI-driven tasks in real-time, such as image registration and reconstruction for tumor tracking in radiotherapy [189].

Alongside neural networks, other ML techniques remain largely used due to their ability to accurately solve classification and regression problems without the need for expensive computational resources [1]. ML can also assist in diagnosing conditions from clinical data, such as myocardial infarction [190,191] and in making differential diagnoses among various findings in neonatal radiographs [192]. For instance, decision trees have demonstrated the ability to predict specific phenotypes from raw genomic data [193], to assign emergency codes based on symptoms during triage in emergency departments [194], and other tasks [195].

Since recently, multi-input AI models can merge and mine the complementary information encoded in omics data, EHRs, imaging data, phenomics, and environmental data of the patient, which represent a current technical challenge [152]. Meta AI’s Imagebind [196] and Google Deepmind’s Perceiver IO [197] represent significant advancements in processing and integrating multimodal data. Other than the flexibility of architecture, other reasons contributed to the large adoption of DL in medical imaging [198,199,200]. Despite these successes, there are also challenges. This section explores both the factors contributing to AI’s success and the concerns surrounding its use.

### 5.1. Open-Source Libraries and Databases

The availability of open-source and free libraries has greatly facilitated the adoption of DL by researchers and clinicians. TensorFlow (https://www.tensorflow.org) and PyTorch (https://pytorch.org), both of which can be run using Python (www.python.org), a high-level and easy-to-learn language, are widely used for developing DL-based segmentation and classification tools. This trend has been highlighted in numerous systematic surveys [1].

Medical Open Network for Artificial Intelligence (MONAI) [201,202] is a project initiated in 2020 by NVIDIA and King’s College London, which has since evolved into the MONAI framework. This open-source framework, built on PyTorch, focuses on DL in healthcare imaging. Its goal is to develop and share best practices for AI in medical imaging, offering state-of-the-art, end-to-end training workflows. MONAI provides researchers with an optimized and standardized approach to creating and evaluating DL models. The framework includes workflows for utilizing domain-specific networks, loss functions, metrics, and optimizers [203].

The research community increasingly tends to share their programming code, often organized as easy-to-run scripts with clear instructions, alongside their datasets. This practice aims to make research studies more reproducible [204,205]. Sharing data, including raw and processed images (with segmentations and annotations) and clinical data, in public repositories is also highly encouraged [206,207,208,209]. Public medical databases, like the Cancer Imaging Archive [210], can be used to train and validate new DL models.

### 5.2. Real World Evaluation

A significant portion of AI tools lacks evidence of efficacy published in peer-reviewed scientific journals [211]. Additionally, the performance achieved during the research phase is often difficult to replicate in clinical settings [212].

In a review of 2021, out of the AI-powered medical devices approved or cleared by the US Food and Drug Administration, only a small number have been tested through prospective randomized controlled trials [213]. Therefore, for AI tools to be integrated into clinical practice, systematic clinical evaluations or trials are necessary.

Many clinical trials were introduced at the end of 2023, as pointed out by a recent review: eighty-six randomized clinical trials for AI-based tools were registered, mostly diagnostic, predominantly as single-center trials [214,215]. The ScreenTrustCAD prospective clinical trial demonstrated that AI alone was non-inferior to double reading by two radiologists in screening mammography. Additionally, combining AI with a single radiologist outperformed double reading by two radiologists, likely due to AI’s high sensitivity in detecting cancer and the ability of consensus readers to improve specificity by dismissing AI-generated false positives [2].

### 5.3. Explainability/Interpretability

A key barrier to the widespread adoption of AI-based tools in medical imaging is that these systems are often viewed as black boxes, making it challenging to understand how they arrive at their decisions [216].

Interpretability and explainability in AI relate to understanding how an AI system arrives at its decisions or outputs [217]. Although these terms are often used interchangeably, interpretability generally refers to either designing models that inherently reveal insights into the patterns they learn during training or analyzing the model to uncover relationships it has identified, such as by examining feature importance [218]. Techniques to enhance interpretability include visual methods like saliency maps and heatmaps, which highlight areas an image-based model considers significant for its predictions [219]. Explainability, in contrast, focuses on making the AI’s decision-making process more understandable and communicable to humans [216,220].

The scientific community is developing explainability frameworks to make AI models more transparent and understandable to humans, leading to “explainable AI” (XAI). Both system developers and end users can obtain, in this way, an idea of the motivations behind the decision provided by an AI system. XAI is essential in DL applications to medical imaging to ensure transparency, accountability, safety, regulatory compliance, and clinical applicability [216].

By providing interpretable explanations for DL model predictions, XAI techniques enhance the trust, acceptance, and effectiveness of AI systems in healthcare, ultimately improving patient outcomes and advancing the field of medical imaging.

### 5.4. Ethical Issues

AI also brings risks and ethical issues, such as the need to ensure fairness, meaning it should not be biased against some group or minority [221]. Bias in AI software may result from unbalanced training data. For instance, due to a severe imbalance in the training data, an AI tool for diagnosing diabetic retinopathy was found to be more accurate for light-skinned subjects than for dark-skinned subjects [222]. Another risk arises from differences between the training data used to develop the algorithm and actual patient data, known as data shift. Changes in patient or disease characteristics or technical parameters (e.g., treatment management and imaging acquisition protocol) over time or across locations can affect the accuracy of AI (input data shift) [223]. Additional risks related to AI include cybersecurity challenges [224]. Implementing AI requires managing these risks through a quality assurance program and quality management system [225]. Within the European Union (EU), the regulatory framework for medical devices is defined by the European Medical Devices Regulation (EU) 2017/745 (EU MDR) and the General Data Protection Regulation (EU) 2016/679 (GDPR), which established criteria for AI implementation [3,226]. Furthermore, the EU has proposed legislation known as “The Artificial Intelligence Act (AI Act)” [227], aiming at creating a unified regulatory and legal structure for AI.

## 6. Conclusions

After a long and tumultuous history, we are in a phase of enthusiasm and promises regarding AI applications to medicine. Fueled by its versatility, impressive results, and the availability of powerful computing resources and open-source libraries, AI is one of the most promising frontiers in medicine. Some medical imaging tasks can be successfully addressed by traditional ML methods like RF, which is less prone to overfitting than DL and more easily interpretable. Various DL architectures can efficiently and accurately perform a range of tasks, including image reconstruction and registration. DL networks have also achieved human-level performance in tasks such as lesion detection, image classification, and segmentation. Additionally, foundation models, pre-trained on a large scale, can be fine-tuned for diverse domains, requiring less training data than training a DL model from scratch.

Indeed, to facilitate the diffusion of AI-based tools in clinical workflows, in addition to the development of increasingly cutting-edge technological solutions that can answer different clinical questions, AI-based systems should be validated in large-scale clinical trials to demonstrate their effectiveness. Additional concerns regarding AI in healthcare must be addressed, including the view of AI tools as ‘black boxes’, which calls for more interpretable and explainable models to earn the trust of both doctors and patients. Ethical issues, such as ensuring fairness and reliability in AI systems, also need careful consideration.

## Figures and Tables

**Figure 1 cancers-16-03702-f001:**
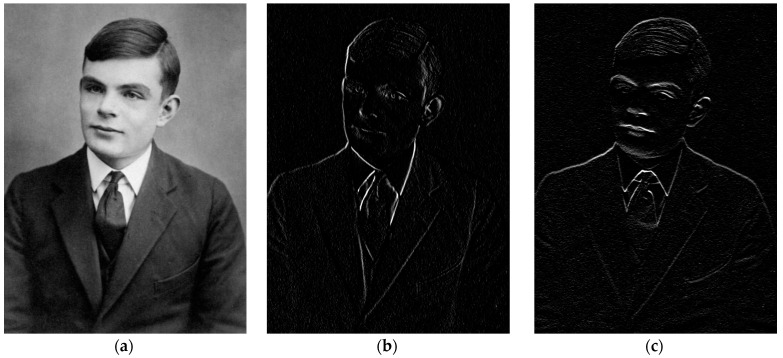
Alan Turing at age 16 (**a**). Source: Archive Centre, King’s College, Cambridge. The Papers of Alan Turing, AMT/K/7/4. The same image after applying a Sobel filter in the x (**b**) and y (**c**) direction.

**Figure 2 cancers-16-03702-f002:**
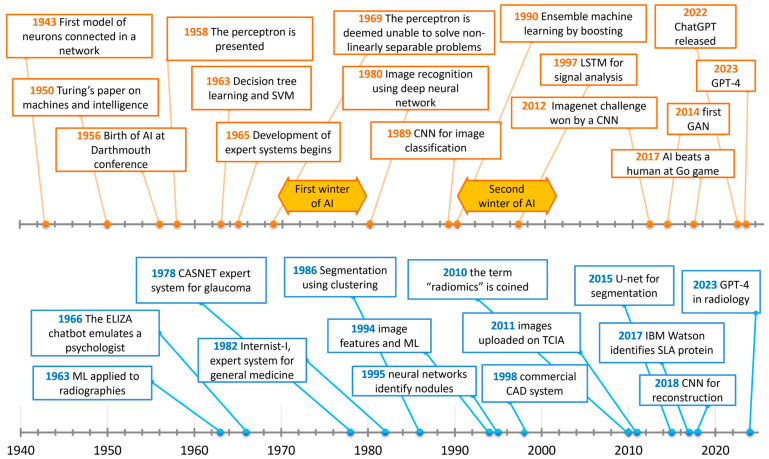
Timeline of AI (orange) and of AI in medicine (blue).

**Figure 3 cancers-16-03702-f003:**
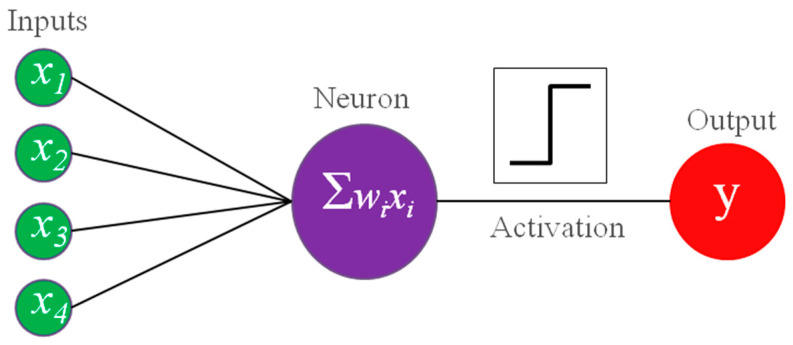
Scheme of the perceptron.

**Figure 4 cancers-16-03702-f004:**
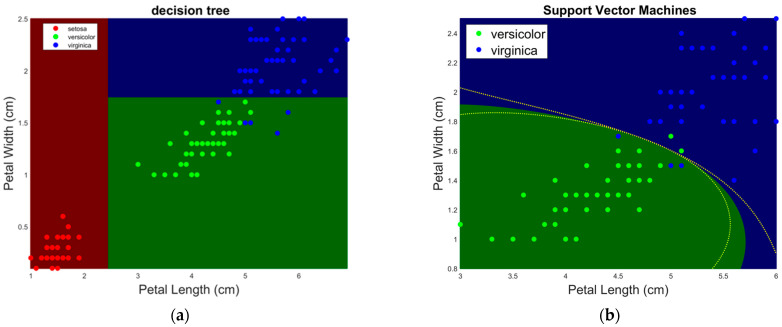
Application of decision trees (**a**) and support vector machines (**b**) learning to the classification of iris flower species from petal width and length. Prediction (areas) and training data (dots) and the resulting decision tree are shown on the left and right sides, respectively.

**Figure 5 cancers-16-03702-f005:**
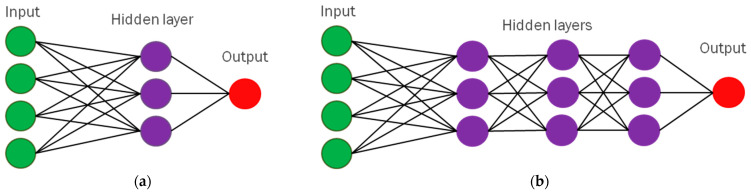
Comparison between single-layer (**a**) and multilayer ANNs (**b**).

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
