# Peer review of "The Evolution of Artificial Intelligence in Medical Imaging: From Computer Science to Machine and Deep Learning"

_cancers, 2024, doi:10.3390/cancers16213702_

Round 1

Reviewer 1 Report

Comments and Suggestions for Authors

Dear Authors,

Thank you for the opportunity to review your valuable paper. This comprehensive review, spanning from the early history of AI to its current applications in the medical field, is impressive and will be an excellent resource for students and researchers alike.

Major comments:

I strongly recommend including sections on self-supervised learning and foundation models. These topics are crucial in the current AI landscape, especially in the medical field where data scarcity is a common issue.

Ex: foundation model

- DOI: 10.1038/s41591-024-02857-3

Example of application

- DOI: 10.1002/2056-4538.12370

(You don't have to cite thoes papers, it is just examples)

Consider discussing how foundation models, trained through self-supervised learning, are revolutionizing medical image analysis, particularly in fields like pathology. Recent developments have enabled significant advancements with limited data. In section 4.4, where you discuss GPT and other language models, it would be beneficial to elaborate on how these models can be fine-tuned (based-on LoRA) for various medical downstream tasks. This is particularly relevant because these LLMs are also considered foundation models.

Minor comments:

Line 163: Change "first winter" to "first AI winter" for clarity.

Line 256: Correct "th ML model" to "the ML model".

Line 276: Capitalize "neocognitron" to "Neocognitron".

Line 529: Correct "he" to "The".

Line 504: Fix typo "trasformer-based" to "transformer-based".

Line 508: Correct spelling from "analize" to "analyze".

Line 562: Correct "implementating" to "implementing".

Overall, this paper is well-written and makes a significant contribution to the field.

I believe it deserves to be published and will be a valuable resource for the scientific community.

Author Response

Comment: Thank you for the opportunity to review your valuable paper. This comprehensive review, spanning from the early history of AI to its current applications in the medical field, is impressive and will be an excellent resource for students and researchers alike.

Response: thank you for your kind response ad review of our manuscript. The revised version was modified accordingly and we hope its improvements make itacceptable for publication. In the revised manuscript, the sections that were modified are highlighted in red for better clarity.

Major comments:

Comment: I strongly recommend including sections on self-supervised learning and foundation models. These topics are crucial in the current AI landscape, especially in the medical field where data scarcity is a common issue. Ex: foundation model:

- DOI: 10.1038/s41591-024-02857-3

Example of application

- DOI: 10.1002/2056-4538.12370

(You don't have to cite those papers, it is just examples)

Consider discussing how foundation models, trained through self-supervised learning, are revolutionizing medical image analysis, particularly in fields like pathology. Recent developments have enabled significant advancements with limited data. In section 4.4, where you discuss GPT and other language models, it would be beneficial to elaborate on how these models can be fine-tuned (based-on LoRA) for various medical downstream tasks. This is particularly relevant because these LLMs are also considered foundation models.

Response: thank you for the kind suggestion. A section (4.5) on Foundation models was added, where we discuss how they can affect pathology and radiology and introduce self-supervised training. The first article suggested was incorporated.

Comment: Minor comments:

Line 163: Change "first winter" to "first AI winter" for clarity.

Line 256: Correct "th ML model" to "the ML model".

Line 276: Capitalize "neocognitron" to "Neocognitron".

Line 529: Correct "he" to "The".

Line 504: Fix typo "trasformer-based" to "transformer-based".

Line 508: Correct spelling from "analize" to "analyze".

Line 562: Correct "implementating" to "implementing".

Overall, this paper is well-written and makes a significant contribution to the field.

I believe it deserves to be published and will be a valuable resource for the scientific community.

Response: all the typos were corrected. Thank you for pointing them.

Reviewer 2 Report

Comments and Suggestions for Authors

After reviewing the manuscript entitled "The evolution of artificial intelligence in medical imaging: From computer science to machine and deep learning", I would like to inform you that this manuscript has covered a scientific discussion, and it seems to be qualitatively It's an acceptable level that surprised me. The research question is well stated in the text and answered correctly. It seems that with a little editing, its quality can be brought to an acceptable level. Therefore, due to the fact that its methodology is acceptable and scientifically correct and it has presented good material, I draw the attention of the authors to the following points so that the revised version of this manuscript can be brought to the highest level. So pay attention to the details.

1" Try to write the abstract more richly. The abstract should cover all the findings.

2: It is better to deal with image processing and disadvantages of traditional methods first in the introduction. Then prove that based on the applications of artificial intelligence in the field of health care, it has been able to cover the challenges and disadvantages of traditional methods in the field of medical image processing. I suggest you mention the wide range of applications of artificial intelligence in health care. It is better to mention studies like the following studies so that the comprehensiveness of artificial intelligence applications can be covered.

"Artificial Intelligence in Drug Discovery and Development against Antimicrobial Resistance: A Narrative Review"

"AI in Nuclear Medical Applications: Challenges and Opportunities"

"Computer-aided diagnosis software for vulvovaginal candidiasis detection from Pap smear images"

Using these sources can enrich your list of references and your mentioned studies.

4. It is better to mention the state of the art applications of deep learning. Pre-trained models that are very helpful for image analysis.

5: Write the conclusion part richer and more complete. This issue can enrich your manuscript.

Comments on the Quality of English Language

Need minor revision. 

Author Response

Comment: After reviewing the manuscript entitled "The evolution of artificial intelligence in medical imaging: From computer science to machine and deep learning", I would like to inform you that this manuscript has covered a scientific discussion, and it seems to be qualitatively It's an acceptable level that surprised me. The research question is well stated in the text and answered correctly. It seems that with a little editing, its quality can be brought to an acceptable level. Therefore, due to the fact that its methodology is acceptable and scientifically correct and it has presented good material, I draw the attention of the authors to the following points so that the revised version of this manuscript can be brought to the highest level. So pay attention to the details.

Response: thank you for your kind response ad review of our manuscript. The manuscript was revised accordingly and we hope its improvements make itacceptable for publication. In the revised manuscript, the sections that were modified are highlighted in red for better clarity.

Comment: 1" Try to write the abstract more richly. The abstract should cover all the findings.

Response: the abstract was expanded to summarize and distil all the relevant findinds and recommendation from the article.

Comment: 2: It is better to deal with image processing and disadvantages of traditional methods first in the introduction. Then prove that based on the applications of artificial intelligence in the field of health care, it has been able to cover the challenges and disadvantages of traditional methods in the field of medical image processing. I suggest you mention the wide range of applications of artificial intelligence in health care. It is better to mention studies like the following studies so that the comprehensiveness of artificial intelligence applications can be covered.

Response: In section 4.2 we mentioned traditional semiautomated methods for lesion segmentation. In the revised abstract and introduction, we mentioned that artificial intelligence has achieved similar performance as humans in the tasks previously performed manually, i.e. segmentation, classification and detection of lesions.

Comment: "Artificial Intelligence in Drug Discovery and Development against Antimicrobial Resistance: A Narrative Review"

 "AI in Nuclear Medical Applications: Challenges and Opportunities"

"Computer-aided diagnosis software for vulvovaginal candidiasis detection from Pap smear images"

 Using these sources can enrich your list of references and your mentioned studies.

Response: we cited the paper about CAD. Thank you for your kind suggestion.

Comment: 4. It is better to mention the state of the art applications of deep learning. Pre-trained models that are very helpful for image analysis.

Response: We have mentioned them in “foundational models”.

Comment: 5: Write the conclusion part richer and more complete. This issue can enrich your manuscript.

Response: thank you for your suggestion. We added sentences about DL for classification, segmentation, and other tasks, and about the potential of foundation models in the introduction.

Reviewer 3 Report

Comments and Suggestions for Authors

This article presents a comprehensive review of the development and applications of artificial intelligence (AI) in medical imaging, covering historical context, major technical advancements, and specific applications in clinical diagnostics. By detailing the evolution from early symbolic AI and machine learning to modern deep learning models, the authors effectively demonstrate how AI has transformed medical imaging analysis and enhanced clinical decision support systems. The article also addresses the challenges AI faces in clinical adoption, such as data bias, model explainability, and ethical considerations, providing a balanced perspective on both the opportunities and limitations of AI in this field.

Suggestions for Improvement:

1. Section on Open-Source Resources: It would be beneficial to add a dedicated section discussing the role of open-source software libraries (e.g., TensorFlow, PyTorch) and open medical imaging databases (e.g., Cancer Imaging Archive) in supporting the development, reproducibility, and verification of AI models in medical imaging. Such a section could also address how open-source resources contribute to reducing costs, fostering collaboration, and ensuring data compliance, which are crucial for advancing AI in this field.

2. Consistency and Typographical Accuracy: Minor typographical errors, such as "he advent of graphics processing unit" (should be "the advent of graphics processing unit"), should be corrected to enhance readability and professionalism. A thorough proofreading would ensure that similar errors are addressed.

3. Discussion of Explainable AI (XAI): The article could benefit from an expanded discussion on the importance of explainable AI in medical imaging, perhaps with examples of current XAI techniques used in clinical applications. This addition would highlight the necessity for model transparency and align well with the ethical and practical challenges mentioned.

4. Clinical Validation and Real-World Impact: While the review provides an excellent overview of AI’s theoretical and technical advancements, a section summarizing successful clinical trials or real-world implementations would strengthen the practical relevance of the article. This could include challenges and best practices for translating research-stage AI tools into routine clinical workflows.

Author Response

Comment: This article presents a comprehensive review of the development and applications of artificial intelligence (AI) in medical imaging, covering historical context, major technical advancements, and specific applications in clinical diagnostics. By detailing the evolution from early symbolic AI and machine learning to modern deep learning models, the authors effectively demonstrate how AI has transformed medical imaging analysis and enhanced clinical decision support systems. The article also addresses the challenges AI faces in clinical adoption, such as data bias, model explainability, and ethical considerations, providing a balanced perspective on both the opportunities and limitations of AI in this field.

 Response: thank you for kind response ad reviewing our manuscript. The manuscript was revised according to the reviewers’ suggestions and we hope it is improved and acceptable for publication. In the revised manuscript, the sections that were modified are highlighted in red for better clarity.

Suggestions for Improvement:

Comment: 1. Section on Open-Source Resources: It would be beneficial to add a dedicated section discussing the role of open-source software libraries (e.g., TensorFlow, PyTorch) and open medical imaging databases (e.g., Cancer Imaging Archive) in supporting the development, reproducibility, and verification of AI models in medical imaging. Such a section could also address how open-source resources contribute to reducing costs, fostering collaboration, and ensuring data compliance, which are crucial for advancing AI in this field.

Response: A section was added to discuss the role of open-source software libraries and medical imaging databases, indicating references and links to the repositories (section 5.1 in the revised manuscript).

Comment: 2. Consistency and Typographical Accuracy: Minor typographical errors, such as "he advent of graphics processing unit" (should be "the advent of graphics processing unit"), should be corrected to enhance readability and professionalism. A thorough proofreading would ensure that similar errors are addressed.

Response: Thank you, the paper has been thoroughly reviewed for typos. 

Comment: 3. Discussion of Explainable AI (XAI): The article could benefit from an expanded discussion on the importance of explainable AI in medical imaging, perhaps with examples of current XAI techniques used in clinical applications. This addition would highlight the necessity for model transparency and align well with the ethical and practical challenges mentioned.

 Response: A section dedicated to Explainable AI was expanded to discuss the relevant role of this concept medical image analysis(section 5.3 in the revised manuscript).

Comment: 4. Clinical Validation and Real-World Impact: While the review provides an excellent overview of AI’s theoretical and technical advancements, a section summarizing successful clinical trials or real-world implementations would strengthen the practical relevance of the article. This could include challenges and best practices for translating research-stage AI tools into routine clinical workflows.

Response: In sec. 5.2 we added several considerations on this topic. In particular, we discussed a recent review focused on a number of clinical trials

Round 2

Reviewer 3 Report

Comments and Suggestions for Authors

The authors have well addressed my concerns in their revision. I thus suggest "Accept".